# Comparison of Clinical Features of Intervertebral Disc Extrusions in English Cocker Spaniels, French Bulldogs and Dachshunds

**DOI:** 10.3390/ani15040602

**Published:** 2025-02-19

**Authors:** Jad Abouzeid, Nick Grapes, Sam Khan, Steven De Decker, Paul Freeman

**Affiliations:** 1Neurology and Neurosurgery, Southfields Veterinary Specialists, Part of Linnaeus Veterinary Limited, Cranes Point, Gardiners Lane South, Basildon SS14 3AP, UK; 2Neurology and Neurosurgery, Davies Veterinary Specialists, Part of Linnaeus Veterinary Limited, Manor Farm Business Park, High Gobion SG5 3HR, UK; nick.grapes@vetspecialists.co.uk; 3Queen’s Veterinary School Hospital, University of Cambridge, Madingley Road, Cambridge CB3 0ES, UK; shk47@cam.ac.uk (S.K.); pf266@cam.ac.uk (P.F.); 4Department of Clinical Science & Services, Royal Veterinary College, Hatfield AL9 7TA, UK; sdedecker@rvc.ac.uk

**Keywords:** intervertebral disc disease, intervertebral disc extrusion (IVDE), spinal cord injury, chondrodystrophic, canine, English cocker spaniel, French bulldog, dachshund

## Abstract

Spinal injury in dogs caused by intervertebral disc extrusions is common in veterinary neurology. Breeds that are termed chondrodystrophic are known to be genetically susceptible to intervertebral disc extrusions. Three popular breeds include the dachshund, French bulldog and English cocker spaniel. There have been studies comparing the dachshund to both other breeds, but the three have yet to be compared together. This study aimed to describe and compare intervertebral disc extrusions in these three breeds. By retrospectively searching the clinical records at two UK-based referral centres over a five-year period, 937 cases of intervertebral disc extrusion were identified pertaining to these three breeds. When comparing age at presentation, French bulldogs presented at the youngest median age, and English cocker spaniels presented at the oldest age. The thoracolumbar region was the most affected anatomical region in all three breeds; French bulldogs had the highest proportion of extrusions in the cervical region and English cocker spaniels had the highest proportion of extrusions in the lumbar region. The clinical features of intervertebral disc extrusion appear to differ between these three breeds.

## 1. Introduction

Intervertebral disc (IVD) degeneration is a normal ageing process that predisposes dogs to developing IVD disease [1,2,3]. It is estimated that IVD disease accounts for almost 21% of neurological cases in domestic dogs in a referral setting [4]. The overall incidence rate of IVD disease across all breeds is reported to be 2% to 3.5% [2,5]. IVD degeneration can lead to many forms of IVD disease including Hansen type I and type II disc herniations and also potentially more traumatic disc diseases, such as hydrated nucleus pulposus extrusions and acute non-compressive nucleus pulposus extrusions, among others [6,7]. These latter forms of disc disease are known to occur in patients with little to no IVD degeneration, however [7]. Hansen type I disc herniation, also known as intervertebral disc extrusion (IVDE), occurs more commonly in smaller chondrodystrophic breeds, although it can also occur in large non-chondrodystrophic breeds [2,3,8]. IVDE is the most common cause of spinal cord injury in dogs [4,7,9] and is the result of a combination of spinal cord compression and contusion caused by the herniation of degenerated and usually calcified nucleus pulposus through a ruptured annulus fibrosus. Clinical signs include spinal hyperaesthesia and neurological deficits and can vary greatly in severity, and the presence of the disease can significantly impact owner-perceived quality of life [10].

Chondrodystrophy is a term used to describe breeds of dog with short limb length [11]. Chondrodystrophic breeds have been shown to be at increased risk of developing IVDE, with dachshunds being largely overrepresented in most studies [1,2,4,12] and having incidence rates of between 19 and 24% [2,3,12]. The thoracolumbar vertebral region is the most common region for IVDE with T13-L1 and T12-13 disc spaces comprising 21% and 17% of cases, respectively, across all breeds [13]. In contrast, IVDE in the cervical region has a reported incidence of 12.9–25.4% [1,3,14]. Recent genetic studies have identified the FGF4 retrogene insertion on chromosomes 12 and 18 as being associated with chondrodystrophy [11]. Its insertion on chromosome 12 has further been shown to be associated with IVDE, with one study finding this genotype to be associated with a >50 times risk for IVDE [15]. Many breeds have now been identified as expressing this gene and thus having a genetic predisposition for IVDE. Amongst these breeds are the dachshund, French bulldog and English cocker spaniel, three increasingly popular breeds [16,17,18]. IVDE is well characterised in the dachshund, making it a common breed for comparison studies. Comparisons and characterisations of IVDE in the dachshund and English cocker spaniel [19] and in the French bulldog and dachshund have been made [20], however, both of these studies only investigated thoracolumbar IVDE distributions. To the authors’ knowledge, no study exists comparing IVDE characteristics along the entire functional vertebral column for these three common chondrodystrophic breeds: the dachshund, French bulldog and English cocker spaniel. Therefore, the aims of the current study were to describe the anatomical distribution of IVDE in these breeds, as well as the clinical features of the disease across each breed, particularly age at presentation. In addition, we aimed to characterise IVDE in these breeds by comparing the relative risk of them experiencing IVDE in each anatomical location compared to one another.

## 2. Materials and Methods

### 2.1. Case Selection and Medical Record Review

The medical records of dogs diagnosed with IVDE between 1 January 2015 and 31 December 2020 at the University of Cambridge’s Queens Veterinary School Hospital (QVSH) and the Royal Veterinary College Queen Mother Hospital for Animals (QMHA) were retrospectively reviewed. The inclusion criteria included dachshunds, French bulldogs and English cocker spaniels which had undergone magnetic resonance imaging (MRI) investigations and had a diagnosis of IVDE confirmed on their imaging report. The diagnosis of IVDE was determined by the presence of compressive extradural material centred over or near the intervertebral disc space on MRI and/or confirmed intraoperatively by the presence of nucleus pulposus material within the vertebral canal causing spinal cord compression. Dogs that experienced more than one extrusion event during the study period were recorded as separate cases if repeat imaging confirmed a new IVDE. Dogs in which multiple IVDE instances were identified, and cases with IVDE in addition to other concurrent spinal pathology, such as a disc protrusion, were also included. The exclusion criteria included cases in which there were incomplete clinical records or where IVDE was not confirmed on the imaging report. Cases with different types of disc herniation such as hydrated nucleus pulposus extrusion or acute non-compressive nucleus pulposus reported as the only type of herniation as the final diagnosis were also excluded. Data recorded included the signalment (breed, sex and age), presenting clinical signs, neurological examination findings and treatment (before referral and post diagnosis where possible). All dogs had undergone a neurological examination by a European College of Veterinary Neurology (ECVN) diplomate or by an ECVN resident working under the direct supervision of an ECVN diplomate. The imaging reports were reviewed to confirm the diagnosis of IVDE and the findings, including the anatomical site(s) of IVDE, were recorded. At both institutions, the imaging reports were produced by a European College of Veterinary Diagnostic Imaging (ECVDI)-certified veterinary radiologist or an ECVDI resident working under the direct supervision of an ECVDI radiologist. MRI was performed using either a 1.5 T permanent magnet (Intera, Philips Medical Systems, Eindhoven, The Netherlands) or a 0.27 T permanent magnet (Esaote VetMR Grande, Genova, Italy) unit. The acquisition parameters and sequences performed for each case varied between each institution; however, standard protocols included at least T2W sagittal and transverse sequences.

### 2.2. Anatomical Region of IVDE and Neurological Grading

Cases were categorised into anatomical regions based on the location of their IVDE site. The anatomical regions were selected to reflect the functional spinal cord segments, thus allowing interpretation to be clinically relevant and reflective of potential neurological findings. Cases were categorised as cervical (C2/3 to C4/C5), cervicothoracic (C5/C6 to T1/T2), thoracolumbar (T2/T3 to L3/L4), lumbosacral (L4/L5 to L7-S1) or multifocal if multiple IVDE instances were identified across regions. Cases with more than one IVDE instance within the same region were categorised in their respective group as a single case; however, the individual sites of the extrusions were recorded. Using the modified Frankel score [21], a neurological grade between 1 and 5 was assigned to each case based on the neurological examination records. Cases were then categorised as ‘ambulatory’ if their neurological grade was 0–2 and ‘non-ambulatory’ if their grade was 3–5.

### 2.3. Statistical Analysis

Data were stored in Microsoft Excel and exported to RStudio (2020) and JASP for statistical analysis. Continuous variables were tested for normality using the Shapiro–Wilk test and normal Q-Q plots and are reported as either mean ± standard deviation (SD) or median and interquartile range (IQR), dependent on their distribution. Proportions and odds ratios are reported with 95% confidence intervals (CIs) where relevant.

## 3. Results

### 3.1. Overall

A total of 937 dogs met the inclusion criteria for the study. The signalment of these including age, sex and breed is shown in Table 1. Of the total number of included dogs, 50% were dachshunds (*n* = 465), 35% were French bulldogs (*n* = 327) and 15% were English cocker spaniels (*n* = 145).

### 3.2. IVDE Site Comparison

Including the dogs that showed evidence of more than one extrusion at the time of diagnosis, a total of 947 IVDE sites were reported. The most common IVDE site for the French bulldog was the C3-C4 intervertebral disc (17%), followed by L2-L3 (13%) and L1-L2 (12%). For the dachshund, the most common location was T12-T13 (27%), followed by T13-L1 (20%). The most common IVDE location for the English cocker spaniel was T13-L1 (12%), followed by L2-L3 (10%). The location distribution of the IVDE site is shown in Figure 1 as a percentage for each respective breed.

### 3.3. IVDE Anatomical Region

Following categorisation into anatomical regions, 30% (95% CI 24–36) of French bulldogs had an instance of IVDE in the cervical region, 4% (95% CI 2–6) in the cervicothoracic region, 54% (95% CI 46–62) in the thoracolumbar region and 12% (95% CI 8–16) in the lumbosacral region. One French bulldog had a multifocal distribution. In the population of dachshunds, 2% (95% CI 1–4) had an instance of IVDE in the cervical region, 2% (95% CI 0–3) in the cervicothoracic region, 92% (95% CI 84–100) in the thoracolumbar region and 3% (95% CI 2–5) in the lumbosacral region. One dachshund had a multifocal distribution. Of English cocker spaniels, 19% (95% CI 12–26) had IVDE in the cervical region, 8% (95% CI 4–13) in the cervicothoracic region, 52% (95% CI 41–64) in the thoracolumbar region and 21% (95% CI 13–28) in the lumbosacral region, and none which were classified as having a multifocal distribution. The anatomical distribution of IVDE following anatomical region categorisation for each breed is shown in Figure 2 as a percentage.

### 3.4. IVDE Anatomical Region Age Comparison

#### 3.4.1. All Regions

The mean age for the population of French bulldogs, dachshunds and English cocker spaniels presenting with IVDE (all regions) is shown in Table 1 and Figure 3. The mean age of presentation differed between all three breeds, with French bulldogs being the youngest (46.1 m ± 16.5 m), dachshunds being older (70.4 m ± 21.6 m) and English cocker spaniels being the oldest (91.5 m ± 30.5 m) (Figure 3).

#### 3.4.2. French Bulldogs

The mean age at presentation for French bulldogs was youngest for thoracolumbar IVDE (42.3 m ± 14.2 m) and oldest for cervicothoracic IVDE (58.5 m ± 24.6 m). The mean ages for cervical and lumbosacral IVDE were 49.5 m (±17.5) and 49.7 m (±15.7), respectively.

#### 3.4.3. Dachshunds

Similarly, the mean age at presentation for dachshunds was youngest in the thoracolumbar (69.5 m ± 21.3 m) and lumbosacral spine (74.3 m ± 24.8 m) and oldest in the cervical (88.7 m ± 20.8 m) and cervicothoracic spine (89.3 m ± 14.6 m).

#### 3.4.4. English Cocker Spaniels

The mean age at presentation for English cocker spaniels was similar across each region, with cervical IVDE presenting at 95.6 m (±30.2 m), cervicothoracic IVDE at 91.0 m (±28.3 m), thoracolumbar IVDE at 90.0 m (±32.3 m) and lumbosacral IVDE at 91.6 m (±27.8 m).

### 3.5. Ambulatory Status Comparison

The proportion of non-ambulatory dogs based on breed and IVDE anatomical location is summarised in Table 2. Across all locations, the highest proportion of dogs presenting as non-ambulatory was found in dachshunds (61.7%; 95% CI 57.2–66.0). In the thoracolumbar region, however, the French bulldog was the breed with the highest proportion of non-ambulatory presentations (76.7%; 95% CI 63.8–89.6).

### 3.6. Odds Ratios and Incidence

The incidence rates for IVDE in each location for each breed are shown in Figure 2. The region with the highest IVDE incidence rate was the thoracolumbar region in all three breeds. However, French bulldogs had greater odds of presenting with cervical IVDE relative to dachshunds (OR 17.92; 95% CI 9.42–34.09), English cocker spaniels (OR 1.90; 95% CI 1.17–3.07) or both the two other breeds combined (OR 6.54; 95% CI 9.42–34.09). Likewise, an increased odds of presenting with thoracolumbar IVDE was identified in dachshunds relative to French bulldogs (OR 10.54; 95% CI 7.01–15.84), English cocker spaniels (OR 11.15; 95% CI 6.94–17.92) and the two other breeds combined (OR 10.72; 95% CI 7.27–15.83). Finally, English cocker spaniels had greater odds of presenting with lumbosacral IVDE relative to dachshunds (OR 7.32; 95% CI 3.86–13.89), French bulldogs (OR 1.93; 95% CI 1.14–3.25) or the two other breeds combined (OR 3.50; 95% CI 2.15–5.69).

## 4. Discussion

The results of the current study suggest that IVDE is likely to occur at an older age in English cocker spaniels than in French bulldogs and dachshunds, and in French bulldogs at a younger age than in the other two breeds. The distribution of IVDE can also be seen to differ between the three breeds; whilst the thoracolumbar region was most affected in all three, the French bulldog experienced a higher proportion of cervical IVDE instances than the other two, whereas in this population of dachshunds, IVDE was almost exclusively confined to the thoracolumbar region.

The median age of IVDE presentation in the French bulldog and dachshund was similar to that previously reported [20,22,23,24,25]. The median age of English cocker spaniels in the current study was slightly older than reported previously [19]. One possible explanation for this is the exclusion of cervical IVDE cases in the previous study [19]. It has been reported that older dogs have a higher incidence of cervical IVDE [3,26] and that dogs with cervical IVDE are significantly older than dogs with thoracolumbar IVDE [27]. In the current study, the English cocker spaniels presenting with cervical and cervicothoracic IVDE had an older median age than those with IVDE in other locations and, as a result, the inclusion of these cases may have contributed to the higher median age that we report.

The reason for the difference in the median age of presentation across these three breeds is unclear. One possible explanation involves genetic differences. As mentioned previously, it is now understood that the increased expression of the CFA12 FGF4 retrogene is associated with chondrodystrophy and an increased risk of IVDE [11,15]. All three breeds included in this study have been reported to have a similarly high frequency of expression of the CFA12 FGF4 retrogene of <0.9 [11]. Despite this, these breeds are repeatedly shown to have a differing age of onset for IVDE, suggesting that additional factors, perhaps yet to be identified, play a role in IVDE. Recently, the DVL2 gene has been shown to have a fixed high prevalence in screw tail breeds such as the French bulldog [28]. It has since been speculated that such additional co-expressional phenotypes may exacerbate FGF4 related pathology via a yet unknown mechanism [22]. Further genetic studies would be required to investigate whether small differences in the frequency of FGF4 retrogene expression or co-expressional phenotypes results in differing patterns of IVDE with regard to the rate of degeneration and anatomical region affected between breeds.

Anatomically, we report an overall incidence of cervical IVDE of 15% and thoracolumbar IVDE of 73%, in agreement with previous reports [1,3,25,29]. This bimodal anatomical distribution of IVDE in all three breeds is supportive of the notion that IVDE occurs mostly within these two anatomical regions [3,23,24]. Again, in agreement with most previous reports, no IVDE occurred between the T1-T2 and T9-T10 intervertebral discs in our study. A recent review of cranial thoracic myelopathies in dogs revealed that only 4.8% of cases were a result of IVDE [30]. The reduced incidence of IVDE in this region is believed to be a result of the increased dorsal reinforcement of the intercapital ligament, which connects the costal heads in this location [3].

The current study revealed that French bulldogs had the highest prevalence of cervical IVDE amongst the three breeds, accounting for 30% of all cases in this breed. Previous reports have indicated that smaller chondrodystrophic breeds such as the dachshund and beagle tend to have a higher prevalence of cervical IVDE when compared to other breeds [3,31,32]. However, in the current study, dachshunds, despite being overrepresented, had a cervical IVDE prevalence of just 2%. Our findings are consistent with more recent reports, which showed a prevalence of cervical IVDE in French bulldogs of 28–29% and in dachshunds of 9% [25,27]. One possible explanation for the higher prevalence of cervical IVDE in French bulldogs and lower prevalence in dachshunds compared to earlier studies might be the changing breed popularity over the last two decades, during which these studies were conducted. In the UK, during the study period covered by this report, the number of registrations of French bulldogs, dachshunds and English cocker spaniels has increased by 508%, 359% and 29%, respectively [16,17,18]. As a result, this dramatic increase in French bulldog and dachshund registrations may correlate with differences in the population characteristics of these breeds, but further work would be required to assess this. One further possible explanation for the differences in prevalence of cervical IVDE in French bulldogs pertains to anatomical differences in the cervical spine. Dorsal paraspinal musculature and intervertebral disc angulation have been reported to be significantly more prominent and steeper, respectively, in dachshunds compared to a non-chondrodystrophic breed (labrador retriever) at certain levels of the cervical spine [33]. In addition, there are reportedly significant differences in these same measurements when French bulldogs are compared to the dachshund [34]. These differences may provide an anatomical and biomechanical factor influencing the development of IVDE in these breeds; however, further studies evaluating these measurements in various breeds, including the English cocker spaniel, are required for the further clarification of the significance of these differences and the incidence of IVDE.

The pathophysiology of IVD degeneration, calcification and extrusion is understood to be multifactorial, with genetics, body conformation, body weight and neuter status shown to have an impact on IVDE occurrence [11,13,15,22,35]. Congenital malformations are not directly associated with IVDE; however, they may be associated with more distant lumbar IVDE [20,36]. As a result, vertebral mechanical stress in dogs with segments of congenital malformation may also contribute to the variability in IVDE distribution in breeds commonly affected with malformations, such as French bulldogs [20]. The prevalence of concurrent vertebral malformations was not assessed in this current study, and so it cannot be determined whether this was the case.

Anecdotally, amongst veterinary surgeons and neurologists, there appears to be a notion that French bulldogs suffer from a more ‘severe’ form of IVDE compared to other breeds. Factors such as an increased recurrence rate or an increased risk of myelomalacia may contribute to this notion [20,37,38,39,40]. IVDE in the thoracolumbar region in French bulldogs has also been reported to present with more longitudinally extensive haemorrhagic extradural material compared to the cervical region, which may also contribute to this notion [25]. Our study found the French bulldog showed the highest proportion of non-ambulatory presentations for the thoracolumbar region, although over all anatomical regions, there was little difference across the breeds. As a result, the findings of this study do not definitively support the notion that French bulldogs suffer from a more severe form of IVDE with regard to neurological grade at the point of presentation compared to the other two breeds. In addition, the severity of the neurological grade at the point of presentation may be difficult to interpret. The time between the onset of clinical signs and presentation was not standardised in the current study, and time-dependent pathological processes such as the development of myelomalacia may affect the neurological grade at the time of presentation. It remains unclear whether French bulldogs have an increased risk of acute deterioration to non-ambulatory status with IVDE compared to other breeds, and further studies comparing them are required to confirm or dispute this claim.

The current study was limited due to only looking at data collected from two UK referral hospitals, which will have introduced referral bias in this population of dogs. As such, some clinical aspects of the present population, such as the ambulatory status, may not be reflective of the wider population of dogs with IVDE. Despite the current study being multi-centre and retrospective in nature, these features were not considered a limitation in the main results, as this would not have influenced the age at presentation and final diagnosis of IVDE location. A further limitation of the current study was the exclusion of IVDE diagnoses made using computed tomography (CT), focusing only on cases with MRI diagnoses. A recent review of the imaging modalities for IVDE summarised that although CT can be reliably used for diagnosis, it may not be appropriate for all breeds of dog [41]. In addition, without the use of contrast studies, even in appropriate breeds such as the dachshund, false negatives are possible [42], and it has been reported that up to 5% of cases with IVDE required additional imaging (CT contrast myelography or MRI) to achieve a diagnosis [43]. MRI is considered the gold standard for IVDE diagnosis [41], and both institutions involved in the current study had access to this imaging modality for emergency cases and at the weekend; hence, the current studly focused only on this modality. As a result, the number of cases excluded is likely minimised by this; however, a financial bias may still have been introduced as CT imaging may have been selected for cases in which the owners had cost concerns, as this imaging modality is often less costly that MRI.

## 5. Conclusions

In conclusion, the current study shows that French bulldogs are more likely to present with IVDE at a younger age than English cocker spaniels and dachshunds and that English cocker spaniels are older than the two other breeds at the time of presentation. Compared to both French bulldogs and English cocker spaniels, the dachshund is least likely to experience cervical or lumbosacral IVDE, respectively, and is overall most likely to be non-ambulatory on presentation.

## Figures and Tables

**Figure 1 animals-15-00602-f001:**
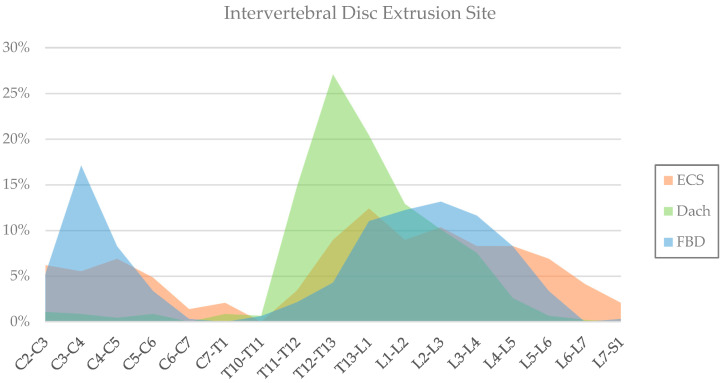
A 2D bar chart showing the anatomical distribution of intervertebral disc extrusion sites for the English cocker spaniel (ECS, in orange), dachshund (Dach, in green) and French bulldog (FBD, in blue) as a percentage within the respective breeds.

**Figure 2 animals-15-00602-f002:**
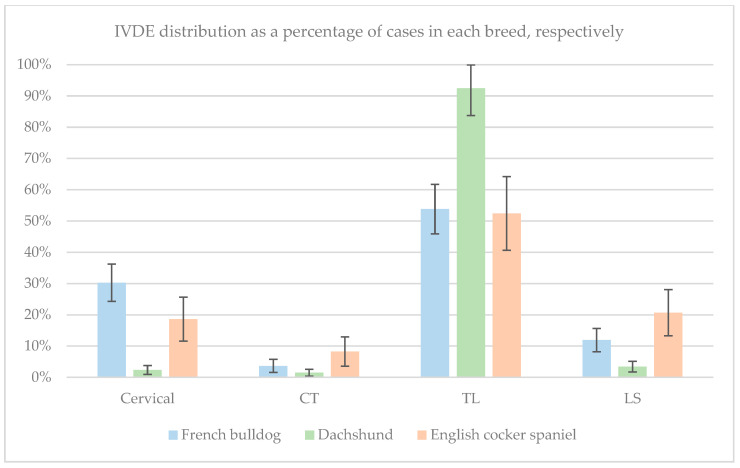
A bar chart showing the distribution of intervertebral disc extrusion sites following categorisation into anatomical regions for English cocker spaniels (orange), dachshunds (green) and French bulldog (blue) as a percentage within the respective breeds. The 95% confidence intervals are displayed as error bars for each respective value. CT, cervicothoracic; TL, thoracolumbar; LS, lumbosacral.

**Figure 3 animals-15-00602-f003:**
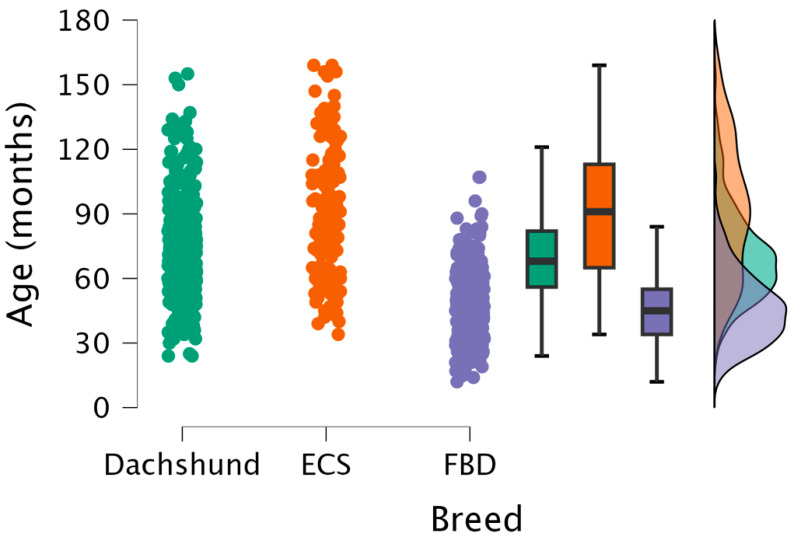
A combined dot, box-and-whisker and raincloud plot of the age (in months) of presentation of IVDE for each breed. ECS, English cocker spaniel; FBD, French bulldog.

**Table 1 animals-15-00602-t001:** Comparisons of the signalment for the included population of French bulldogs, dachshunds and English cocker spaniels. Values represent the number of dogs with the percentages shown in brackets (%). The mean age is shown in months, with the standard deviation also shown (±).

Sex and Age		French Bulldog (*n* = 327)	Dachshund (*n* = 465)	English Cocker Spaniel (*n* = 145)
Sex (%)	Female entire	29 (8.9)	51 (11.0)	4 (2.8)
	Female neutered	83 (25.4)	149 (32.0)	52 (35.9)
	Male entire	71 (21.7)	76 (16.3)	25 (17.2)
	Male neutered	144 (44.0)	189 (40.6)	64 (44.1)
Age [months]	Mean (± standard deviation)	46.1 (±16.5)	70.4 (±21.6)	91.5 (±30.5)

**Table 2 animals-15-00602-t002:** Comparisons of the number and percentage of dogs with IVDE presenting as non-ambulatory for each breed and each anatomical region. The percentage is given for each anatomical region, and the number of dogs that were non-ambulatory for each breed and each region is provided in brackets.

Anatomical Region	French Bulldog (*n* = 327)	Dachshund (*n* = 465)	English Cocker Spaniel (*n* = 145)	All Breeds (*n* = 937)
All regions	51.7% (169/327)	61.7% (287/465)	46.9% (68/145)	55.9% (524/937)
Cervical	15.2% (15/99)	36.4% (4/11)	40.7% (11/27)	21.9% (30/137)
Cervicothoracic	0% (0/12)	0% (0/7)	16.7% (2/12)	6.5% (2/31)
Thoracolumbar	76.7% (135/176)	63.3% (272/430)	61.8% (47/76)	66.6% (454/682)
Lumbosacral	48.7% (19/39)	68.7% (11/16)	26.7% (8/30)	44.7% (38/85)
Multifocal	0% (0/1)	0% (0/1)	na/na	0% (0/2)

## Data Availability

The raw data obtained supporting the conclusions of this publication can be made available by the main author upon request.

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
