# Peer review of "Comparison of Clinical Features of Intervertebral Disc Extrusions in English Cocker Spaniels, French Bulldogs and Dachshunds"

_animals, 2025, doi:10.3390/ani15040602_

Round 1
Reviewer 1 Report
Comments and Suggestions for Authors
Dear authors,
My compliments on this well written manuscript with clear figures and tables. I only have a few comments:
Line 59: compared to body length
Table 1: years instead of months would be more convenient.
Line 236: in a paper published by Park et al in 2009, the FGF4 retrogene insertion was found in Dachshunds but not in French Bulldogs (English cocker spaniel was not included but the American cocker Spaniel was not affected). This might explain the higher incidence in Dachshunds à https://pmc.ncbi.nlm.nih.gov/articles/PMC2748762/
Author Response
1. Summary
Dear reviewer, thank you very much for taking the time to review this manuscript and for your kind complements. Please find the detailed responses below and the corresponding revisions/corrections highlighted/in track changes in the re-submitted files. I hope they are to your satisfaction.
2. Point-by-point response to Comments and Suggestions for Authors
Comments 1: Line 59: compared to body length
Response 1: Thank you for this suggestion. However, I believe the definition of chondrodysplasia refers only to shortened limb phenotype with no reference to body length, although naturally I agree, this will mean a relative difference. Given the definition only stating limb length, I believe the current phrasing best represents the accepted definition. Some examples of publications supporting this are found in the following references: DOI: 10.1126/science.1173275, https://doi.org/10.3390/genes10060435 and https://doi.org/10.1073/pnas.1709082114. I hope you agree with this decision.
Comments 2: Table 1: years instead of months would be more convenient.
Response 2: Thank you for this suggestion. Considering the varying age of presentation of IVDE, we decided to represent the age in months allows for a better understanding and analysis of the age-related differences found in this study in smaller increments. In addition, for future comparative studies, this decision allows for easy comparison as it reduced ambiguity that maybe found with reporting ages in years. For example, a case which was 4 years and 4 months would be represented as 4.33y and a case which was 4 years and 3 months would be represented as 4.25y. When rounding these ages to one decimal point, they would both be represented by 4.3y. I hope you agree and are stratified with the decision to decision to report the ages in the current study in months.
Comment 3: Line 236: in a paper published by Park et al in 2009, the FGF4 retrogene insertion was found in Dachshunds but not in French Bulldogs (English cocker spaniel was not included but the American cocker Spaniel was not affected). This might explain the higher incidence in Dachshunds à https://pmc.ncbi.nlm.nih.gov/articles/PMC2748762/
Response 3: Thank you for the comment and for supplying this reference, it was very interesting to read. As you mentioned, this study described the FGF4 retrogene in dachshunds but not the other two breeds pertaining to this current study. However, more recent genetic studies have included all three breeds. This review from Dickson and Bannasch summaries the genetics nicely and described the FGF4 retrogene insertion in all three of the breeds included in this study, https://doi.org/10.3389/fvets.2020.00431. In table 1 of this review, the allele frequencies of the FGF4 retrogene for over 50 breeds are displayed. In this table, the dachshund, French bulldog and English cocker spaniel are reported to have very similar CFA12 FGF4 retrogene frequency of 0.97, 0.94 and 0.96 respectively. The Dachshund frequency is higher however, they were overrepresented and so may not an accurate explanation alone for the differences in IVDE incidence found in these breeds. I hope you are satisfied with this explanation and please note this study has been refenced in the current article.

Reviewer 2 Report
Comments and Suggestions for Authors
The manuscript is very well written and presents the results clearly.
The data shown are of clinical value and open the door to future research that may explain the cause of the different locations of the lesion between breeds.
In line 222 you say that “the French bulldog experience a higher proportion of cervical IVDE’s than the other two breeds”. Could this be caused by the characteristic anatomy of the muscles in the cervical region, which are shorter and more powerful? Could this cause stress on the cervical vertebrae, especially those in the caudal cervical region?
In Line 253 you say that “no IVDE’s occurred between the T1-T2 and the T9-T10 intervertebral disc”. As you write in line 256, this may be due to the presence of the intercapital ligament. I must add to this that when this ligament is dissected in the dissection room, the ligament is very thick and powerful.
In line 254 you say: “A recent review of cranial myelopathies in dogs…”. When you say “cranial”, I understand you are referring to thoracic because you are talking about the intercapital ligament.
Author Response
1. Summary
Dear reviewer, thank you very much for taking the time to review this manuscript and for your kind complements. Please find the detailed responses below and the corresponding revisions/corrections highlighted/in track changes in the re-submitted files. I hope they are to your satisfaction.
2. Point-by-point response to Comments and Suggestions for Authors
Comments 1: In line 222 you say that “the French bulldog experience a higher proportion of cervical IVDE’s than the other two breeds”. Could this be caused by the characteristic anatomy of the muscles in the cervical region, which are shorter and more powerful? Could this cause stress on the cervical vertebrae, especially those in the caudal cervical region?
Response 1: Thank you for this comment and bringing it to my attention. Upon review, there have been studies assessing cervical paraspinal musculature. One such study comparing the differences in in a chondrodystrophic breed (dachshund) and non-chondrodystrophic breed (labrador retriever) which did find a significant difference between them and the authors of this paper suggest that this difference may affect the anatomical and biomechanical factors in the development of intervertebral disc disease (https://doi.org/10.3389/fvets.2020.577394). Similarly, another study examined the paraspinal musculature of French bulldogs with and without intervertebral disc disease using the imaging protocols of the aforementioned study allowing for additional comparisons of French bulldogs to dachshunds and the labrador retriever (https://doi.org/10.3389/fvets.2021.705632). These studies do support the notion that there is indeed anatomical differences in the paraspinal musculature (and also intervertebral disc angulation) between dachshunds and French bulldogs which may play a role in the development of intervertebral disc disease. However, no significant differences in muscle height ratios were found between the breeds in the caudal cervical spine, but interestingly there was a significant difference in the cranial cervical spine with the French bulldog having significantly less prominent dorsal paraspinal musculature than the dachshund at the level of C2-3 and C3-4. As a result of these studies, I believe, as the authors suggest, the differences in these paraspinal musculature may contribute to factors which may influence the development of IVDD. A short addition to the paragraph discussing the differences in cervical IVDE prevalence in French bulldogs has been added on page 8, lines 278 -289 reflecting these studies which we hope you are satisfied with.
Comments 2: In Line 253 you say that “no IVDE’s occurred between the T1-T2 and the T9-T10 intervertebral disc”. As you write in line 256, this may be due to the presence of the intercapital ligament. I must add to this that when this ligament is dissected in the dissection room, the ligament is very thick and powerful.
Response 2: Thank you for this comment and for sharing this information. That is very interesting to known and it helps to explain/understand one possible reason for the lack of IVDE prevalence in this region of the spine.
Comment 3: In line 254 you say: “A recent review of cranial myelopathies in dogs…”. When you say “cranial”, I understand you are referring to thoracic because you are talking about the intercapital ligament.
Response 3: Thank you for the comment and for pointing this out. Yes, you are correct in your understanding and so this line has been amended to now reads ‘A recent review of cranial thoracic myelopathies in dogs…’on page 8 (line 260).

Reviewer 3 Report
Comments and Suggestions for Authors
Title: Comparison of clinical features of intervertebral disc extrusions in English cocker spaniels, French bulldogs and Dachshunds
General comments
The authors reported an interesting retrospective observational study on aim to describe and compare the clinical features and anatomical distribution of intervertebral disc extrusions between English cocker spaniel, French bulldog and Dachshund. However, some details must be addressed before publication. See corrections below.
Specific Comments
Introduction
-Change to ..”21% of neurological cases in domestic dogs” (line 46)
-Change to..“Hansen type I and type II”…(line 49)
-Change to..”Clinical signs include spinal pain (or Spinal Hyperesthesia) (line 56)
-Change to the “thoracolumbar vertebral region”…(line 62)
Material and Methods
-Please include a more specific definition of the inclusion criteria for IVDE. This include also ANNPE (acute non-compressive nucleus pulposus extrusion) and/or HNPE (acute compressive hydrated nucleus pulposus extrusion)? (line 94).
May be a definition based on ACVIM consensus statement on diagnosis or similar article.
Results
Table 1: Complete with “ Comparison of the signalment of French bulldog, Dacshund and English cocker spaniel with intervertebral disc extrusion population under study..”(or similar)
Figure 1: Change to “anatomic distribution”..
Dachshunds is presented with capital and sometimes not.
Discussion
Change to “A recent review of cranial thoracic myelopathies….” (line 254)
Discuss the possible influence of vertebral body morphology or morphometric (morphological vertebral features) for the higher prevalence of cervical IVDE in French bulldogs compared with the other breeds (line 265-266).
Discuss that severity of neurological signs at the point of presentation may be a non standard period of time for the reported cases since IVDE pathological associated lesions may varied during time (line 282-295)
Author Response
1. Summary
Dear reviewer, thank you very much for taking the time to review this manuscript and for your kind complements. Please find the detailed responses below and the corresponding revisions/corrections highlighted/in track changes in the re-submitted files. I hope they are to your satisfaction.
2. Point-by-point response to Comments and Suggestions for Authors
Comments 1: Introduction
-Change to ..”21% of neurological cases in domestic dogs” (line 46)
-Change to..“Hansen type I and type II”…(line 49)
-Change to..”Clinical signs include spinal pain (or Spinal Hyperesthesia) (line 56)
-Change to the “thoracolumbar vertebral region”…(line 62)
Response 1: Thank you for these recommended changes. The following changes have been implemented in the revised manuscript.
- ‘21% of canine neurological cases’ having been changed to read ‘21% of neurological cases in domestic dogs…’ (line 47).
-‘Hansen type 1 and type 2’ has been corrected to ‘Hansen type I and type II …’ (line 49)
-‘Clinical signs include spinal pain..’ has been changed to ‘clinical signs include spinal hyperesthesia…’ (line 57)
- ‘thoracolumbar region…’ has been changed to ‘thoracolumbar vertebral region…’ (line 63)
Comments 2: Material and Methods
-Please include a more specific definition of the inclusion criteria for IVDE. This include also ANNPE (acute non-compressive nucleus pulposus extrusion) and/or HNPE (acute compressive hydrated nucleus pulposus extrusion)? (line 94).
May be a definition based on ACVIM consensus statement on diagnosis or similar article.
Response 2: Thank you for this suggestion. A more detailed description of the inclusion criteria has now been included on page 2 (lines 92-96) and the exclusion of ANNPE and HNPE has now been specified in page 3 (lines 101-103). I hope these corrections are to your satisfaction.
Comments 3: Results
Table 1: Complete with “ Comparison of the signalment of French bulldog, Dacshund and English cocker spaniel with intervertebral disc extrusion population under study..”(or similar)
Figure 1: Change to “anatomic distribution”..
Dachshunds is presented with capital and sometimes not.
Response 3: thank you for these suggestions. The following corrections have been implemented in the revised manuscript.
- ‘Comparison of the signalment of the included population of dogs.’ Has been corrected to 'Comparisons of the signalment for the included population of French bulldogs, dachshunds and English cocker spaniels.’ (line 142)
- ‘bar chart showing the distribution…’ has been changes to ‘bar chart showing the anatomical distribution...’ as suggested (line 155).
- Within the body of text, ‘dachshund’ is presented with a lower case and when in figures or tables as a standalone name is presented with a capital letter. We believe this is appropriate.
Comment 4: Discussion
Change to “A recent review of cranial thoracic myelopathies….” (line 254)
Discuss the possible influence of vertebral body morphology or morphometric (morphological vertebral features) for the higher prevalence of cervical IVDE in French bulldogs compared with the other breeds (line 265-266).
Discuss that severity of neurological signs at the point of presentation may be a non standard period of time for the reported cases since IVDE pathological associated lesions may varied during time (line 282-295)
Response 4: Thank you again for these recommendations. The following changes have been implemented in the revised manuscript and I hope these are to your satisfaction.
- ‘A recent review of cranial myelopathies..’ has been changed to ‘A recent review of cranial thoracic myelopathies..’ on page 8 (line 260).
- A new paragraph has been added to discuss the possible influence of paravertebral musculature and angle of the intervertebral disc on the prevalence if cervical IVDE in French Bulldogs compared to other breeds on page 8 (line 278-289). We hope this is an adequate discussion of your recommended point.
- A discussion of possible factors influencing the neurological grade at the time of presentation and the limitations in this metric has now been included on page 9 (lines 310-314). We hope this appropriately explains the limitations of this feature in interpreting the neurological grade at certain time points.

Reviewer 4 Report
Comments and Suggestions for Authors
Thank you for your paper.
It is well written and easy to read.
I only have two comments.
Line 85:
Do you include or exclude cases with hydrated nucleus pulposus extrusion (hydrated nucleus pulposus extrusion; acute non-compressive nucleus pulposus extrusion)? I am surprised that you have no cases with these types of disc herniation.
You didn’t discuss about this category in your introduction also.
Line 290:
It would be interesting to assess the percentage of medullary compression and neurologic status. The velocity of the disc herniation is a component difficult to assess. However, can you add a comment also about the velocity of the hernia is maybe greater in bulldogs.
Author Response
1. Summary
Dear reviewer, thank you very much for taking the time to review this manuscript and for your kind complements. Please find the detailed responses below and the corresponding revisions/corrections highlighted/in track changes in the re-submitted files. I hope they are to your satisfaction.
2. Point-by-point response to Comments and Suggestions for Authors
Comments 1: Line 85: Do you include or exclude cases with hydrated nucleus pulposus extrusion (hydrated nucleus pulposus extrusion; acute non-compressive nucleus pulposus extrusion)? I am surprised that you have no cases with these types of disc herniation.
You didn’t discuss about this category in your introduction also.
Response 1: Thank you for your question. We did not include other types of herniation (such as Type II, HNPE, ANNPE). The current study focused only on interverbal disc extrusions (Hanson type I herniations) as this is the type of herniation has been linked to the FGF4 gene discussed in the introduction and discussion and so this type of herniation is suspected to have different/additional underlying pathological factors aforementioned type of herniations. Lines 50-51 on page 2 have been amended in the introduction to reflect these more relevant types of disc herniation which we hope you are satisfied with.
Comments 2: It would be interesting to assess the percentage of medullary compression and neurologic status. The velocity of the disc herniation is a component difficult to assess. However, can you add a comment also about the velocity of the hernia is maybe greater in bulldogs.
Response 2: Thank you for your comment. I agree, it would be interesting to assess the differences in medullary compression and correlate this to neurological status, however this was out of the scope of focus for the current study. However, we hope this current study which discusses the differences in ambulatory status between these breeds will encourage further research seeking to evaluate this. As you mention, the velocity of disc herniation is difficult to assess and imaging features such as concurrent intramedullary T2 hyperintensities may suggest spinal cord contusions which could be supportive of a high velocity disc herniation. However, given the limitation in assessing intramedullary hyperintensities on MRI alone without histological assessment, it would be impossible to differentiate such contusive lesions from oedema and myelomalacia for example. One of the included references in the current study (https://doi.org/10.3389/fvets.2023.1230280) hypothesizes that a higher velocity extrusion in the thoracolumbar spine may be a contributing factor of the differences in clinical severity of cervical vs thoracolumbar IVDE in the French bulldog, however no definitive assessment in disc herniation velocity has been described in the literature. As a result of the limitations in determining the velocity of disc herniation, we believe it best to not speculate on this feature of the included herniations and we hope you agree.
